# Exploring Dengue Dynamics: A Multi-Scale Analysis of Spatio-Temporal Trends in Ibagué, Colombia

**DOI:** 10.3390/v16060906

**Published:** 2024-06-03

**Authors:** Julian Otero, Alejandra Tabares, Mauricio Santos-Vega

**Affiliations:** 1Centro Para los Objetivos de Desarrollo Sostenible, Universidad de Los Andes, Bogotá 111711, Colombia; 2Grupo Biología Matemática y Computacional (BIOMAC), Universidad de Los Andes, Bogotá 111711, Colombia; om.santos@uniandes.edu.co; 3Departamento de Ingeniería Industrial, Universidad de los Andes, Bogotá 111711, Colombia; a.tabaresp@uniandes.edu.co; 4Departamento de Ciencias Biológicas, Universidad de Los Andes, Bogotá 111711, Colombia

**Keywords:** dengue, spatio-temporal analysis, geographically weighted regression, integrated nested Laplace approximation, spatial aggregation levels

## Abstract

Our study examines how dengue fever incidence is associated with spatial (demographic and socioeconomic) alongside temporal (environmental) factors at multiple scales in the city of Ibagué, located in the Andean region of Colombia. We used the dengue incidence in Ibagué from 2013 to 2018 to examine the associations with climate, socioeconomic, and demographic factors from the national census and satellite imagery at four levels of local spatial aggregation. We used geographically weighted regression (GWR) to identify the relevant socioeconomic and demographic predictors, and we then integrated them with environmental variables into hierarchical models using integrated nested Laplace approximation (INLA) to analyze the spatio-temporal interactions. Our findings show a significant effect of spatial variables across the different levels of aggregation, including human population density, gas and sewage connection, percentage of woman and children, and percentage of population with a higher education degree. Lagged temporal variables displayed consistent patterns across all levels of spatial aggregation, with higher temperatures and lower precipitation at short lags showing an increase in the relative risk (RR). A comparative evaluation of the models at different levels of aggregation revealed that, while higher aggregation levels often yield a better overall model fit, finer levels offer more detailed insights into the localized impacts of socioeconomic and demographic variables on dengue incidence. Our results underscore the importance of considering macro and micro-level factors in epidemiological modeling, and they highlight the potential for targeted public health interventions based on localized risk factor analyses. Notably, the intermediate levels emerged as the most informative, thereby balancing spatial heterogeneity and case distribution density, as well as providing a robust framework for understanding the spatial determinants of dengue.

## 1. Introduction

Dengue fever, a mosquito-borne viral disease [1], has become a critical public health issue globally, particularly in tropical and subtropical regions [2,3]. The country of Colombia, located in the tropical region of Latin America, has reported the highest dengue case fatality rate in the continent, and it has experienced four major outbreaks in the last two decades, occurring in 2010, 2013, 2019, and 2023 [4,5]. Ibagué, is a city in the department of Tolima, Colombia, and it exemplifies an urban area significantly affected by dengue, with its incidence rising notably over the past years [6]. Ibague’s rapid urbanization over the past two decades has led to densely populated, low-income neighborhoods that often lack regular access to water and adequate infrastructure. These socioeconomic conditions have been linked to higher dengue incidence and mosquito populations [7,8,9]. Additionally, the city’s specific environmental characteristics, including its elevation (1225 m above sea level) and mean annual temperature (24 °C), create a conducive environment for mosquitoes of the *Aedes* genus, including *Aedes aegypti* and *Aedes albopictus,* thus, further exacerbating the situation [10,11,12].

Dengue virus transmission follows the mosquitoes’ reproductive and gonotrophic cycles. Female mosquitoes become infected after biting a viremic individual, thereby spreading the disease among humans after each blood meal. Many factors, including urbanization, demographic changes, and environmental conditions, influence this complex dynamic [13,14]. Prior research has demonstrated the critical role of socioeconomic and demographic variables in the spread of dengue [15,16]. However, a significant gap still needs to be addressed in understanding the interaction of these variables at different urban scales, namely levels of aggregation, particularly in rapidly urbanizing cities in developing countries [17].

Understanding the effect of spatial proximity on disease transmission in urban settings is essential to identify vulnerable areas and populations [18,19,20,21,22]. Considering neighborhood structures for internal interactions has proven to be beneficial in the analysis of demographics, socioeconomics, and climate while evaluating their influence over dengue transmission and incidence [23,24,25]. This approach has allowed authors to find correlation with factors such as human population density, access to tap water, and hydrometeorological phenomena [26,27].

Specifically in Colombia, certain authors have implemented a spatial methodology for dengue in the city of Cali using a space–time conditional autoregressive model [28], which was conducted on a neighborhood level. The results showed that lagged weather variables could help to identify when the peaks in the risk of transmission occur. Additionally, they proved that dengue infections are not exclusive to poor areas, and the risk of infection is related to spatial and temporal distribution. The proposed aggregation level of neighborhoods offered sparse data observations with clear socioeconomic and demographic trends.

While the importance of spatial and temporal variables in dengue transmission is recognized [29], limited research has been conducted on integrating these factors at different levels of urban spatial aggregation. This study aims to bridge this gap by leveraging detailed demographic and socioeconomic data from the census, which are provided by the National Administrative Department of Statistics (Departamento Administrativo Nacional de Estadística—DANE), as well as the environmental variables derived from satellite imagery and previous studies. We seek to unravel the spatial and temporal dynamics influencing dengue incidence in Ibagué between 2013 and 2018, thereby examining these factors across four levels of spatial aggregation—Manzanas, Secciones, Sectores, and Comunas [30]. We introduce a novel approach by employing geographically weighted regression (GWR) to isolate key socio-economic and demographic predictors at varying spatial scales [31]. Additionally, the use of INLA models allows for an in-depth examination of spatio-temporal correlations and their posterior distributions [32], thus offering new insights into the localized dynamics of dengue transmission.

The methodological innovation of this study lies in its tripartite modeling strategy, encompassing spatial, temporal, and combined spatio-temporal models at each level of spatial aggregation. This approach allows for a comprehensive analysis of the varying impacts of different factors on dengue incidence, thus providing a nuanced understanding of the disease’s transmission dynamics in an urban setting. Our findings will contribute significantly to public health, particularly in developing targeted dengue control strategies [33]. This study’s framework also offers a valuable model for similar epidemiological investigations in other urban settings, thus enhancing our understanding of vector-borne diseases in global urbanization trends.

## 2. Materials and Methods

### 2.1. Study Site and Data

Ibagué is the most populated city in the department of Tolima, with an estimated population of 541,101 people in 2020 [34]. According to DANE, the city’s urban area is divided into Comunas (communes), Sectores (sectors), Secciones (sections), and Manzanas (blocks), which will be used as the levels of spatial aggregation in this study. Further details about the city’s location are presented in Appendix A, and information of the spatial levels of aggregation is contained in Appendix A.

The minimum aggregation level is at the level of Manzanas, which contains one block each, and the most aggregated level is the level of Comunas, each containing between 169 and 862 blocks. Groups of Manzanas create the rest of the levels, as can be observed in Appendix A. The demographic and socioeconomic details were obtained at all four levels from the National Geostatistical Framework (Marco Geoestadístico Nacional) and the National Population and Dwelling Census (Censo Nacional de Población y Vivienda), which were compiled in 2018 [35]. All of the available socioeconomic and demographic predictors from the census were included in the preliminary analysis.

Socioeconomic predictors included variables related to strata, water access and disposal, access to gas, garbage pick-up services, and internet connection, which allows one to identify lower income and utility access areas inside the city. Demographic variables divide the population according to age, gender, and educational level. Finally, environmental variables were obtained from satellite images via the Google Earth Engine using MODIS11A1 for mean temperature [36], MODIS13Q1 for Normalized Difference Vegetation Index (NDVI) [37], CHIRPS PENTAD for total precipitation [38], as well as from previous studies on the city for wet days and hot days (over 32 °C) [39]. The variables, their description, and the spatial resolution are shown in Appendix A.

Data on dengue cases was obtained from the dataset provided by the local government. Out of the total 17,707 dengue cases recorded from 2013 to 2018, 16,183 were included according to the spatial levels of the city. The remaining cases had no readable address or were reported in the city’s rural area. The years 2013 and 2015 showed the most extensive outbreaks with 5383 and 4885 cases, respectively, with another outbreak following during the first months of 2016. Notably, the number of cases in 2017 and 2018 was significantly lower, with each year reporting fewer than 1000 cases (Figure 1A). Finally, the spatial distribution of dengue cases varied heavily throughout the years, only being consistently high around the central area of the city (Figure 1B).

### 2.2. Methodology

#### 2.2.1. Variable Selection and Transformation

To demonstrate the association between the variables and dengue incidence, as well as reduce the dimensionality of the spatial variables, we used GWR through the package *GWModel* in R [40,41]. This methodology allows for the creation of different local regressions using the Ordinary Least Squares (OLS) method for every spatial feature, thereby adding a weight parameter that was obtained from the distance between the geometries and a calculated bandwidth [42,43]. This analysis was performed for all the available socioeconomic and demographic predictors obtained from the national census on every level of aggregation. Only the variables that were shown to be significant at, at least, one global regression at any level of aggregation were used as the final predictors for the inference model.

We also used a wavelet coherence analysis to confirm the existing correlations between dengue cases and temporal variables. This was used to analyze non-stationary time series. The methodology implements a decomposition between time and frequency using a windowed Fourier transform, which allows for local time-frequency properties while adjusting for high- and low-frequency structures [44]. The wavelet coherence was computed for the aggregated environmental variables in the whole city. The results were evaluated graphically using the *biwavelet* library in R, thereby keeping the variables that showed high correlation and significance in the wavelet coherence analysis as the final predictors [45]. All temporal predicted values were included since they displayed high correlation with dengue cases, as shown in Appendix A.

The temporal predictors were later lagged using a distributed lag nonlinear model (DLNM). This methodology considers the delayed effects and nonlinear relationships between dengue and environmental time-dependent variables [46]. The package *dlnm* was used on R [47], thereby obtaining, as a result, a matrix that accounts for the nonlinear exposure and a delayed effect.

#### 2.2.2. Model Fitting

The latent marginal distribution of the chosen predictors was approximated using INLA, wherein the spatio-temporal influence of these predictors was considered. This approach relies on Latent Gaussian Models (LGM), specifically, a Latent Gaussian Markov Random Field with a sparse and factorizable precision matrix. Such a structure enables numeric approximations, leading to quicker outcomes than conventional LGM techniques like Markov Chain Monte Carlo [32].

We fitted a negative binomial model to the number of cases $Y_{st}$ for each spatial unit s at a given time t, which resulted in obtaining an estimated mean of $\mu_{st}$ and a dispersion parameter ϕ. This model allows us to account for the overdispersion in the number of cases at each scale. Our link function considers $\mu_{st}$ from the population $p_{st}$ and the monthly incidence at the same location and time $\left(\rho_{st}\right)$. The model is depicted in Equations (1) to (3).
(1)$$Y_{st}|\mu_{st}\sim NB(\mu_{st},\phi ),$$
(2)$$\log\left(\mu_{st}\right)=\log\left(p_{st}\right)+log(\rho_{st}),$$
(3)$$\log\left(\rho_{st}\right)=X^{T}\beta +\gamma_{st}+\eta_{st}.$$

We also included two random effects that were considered to account for unobserved variability. An unstructured random effect for seasonal autocorrelation, regarding possible relationships in time for each structure along the months, was determined to be cyclic over a six-year analysis. We also included a second structured random effect, which encompassed the spatial autocorrelation between the neighborhoods during the years regarding interconnection, interventions, herd immunity, etc. [27]. The incidence was calculated from the fixed effects $X^{T}$ and the following two random effects: $\gamma_{st}$ for the unstructured effect following a random walk and $\eta_{st}$ for the structured one using a Besag–York–Molliè model [48]. Precision priors were defined from the precision parameter $P\left(\sigma >0.5\right)=0.01$.

An adaptative strategy was selected as it is considered the best fit for Gaussian and simplified Laplace approximations. Finally, the hyperparameter posterior distributions were calculated with a central composite design as it offers the best tradeoff between precision and computational time among the possible strategies implemented in the R library *INLA* [32,49,50].

Three models were adjusted per level: one containing only spatial variables (socioeconomic and demographic), a second containing only temporal variables (environmental), and a third containing both. This enabled us to distinguish between the effects of the spatial and temporal covariates, while assessing whether incorporating both yielded a more informative model. A comparison was performed for the models at each level using the deviance information criterion (DIC) and widely applicable information criterion (WAIC).

#### 2.2.3. Aggregation Level Comparison

From the selected best-fitting models at each level of aggregation, fitted values were obtained from the marginal posterior distributions of the selected models at each level of aggregation. These values were then compared with the observed data over the studied period to determine the best fitting model at each level by comparing the root mean squared error (RMSE).

## 3. Results

### 3.1. Variable Selection

The GWR analysis revealed a significant association for seven critical variables at a minimum of one spatial level over different years (Figure 2). These variables encompassed critical aspects of the socio-economic landscape, including population density (Density), sewage connection (Sewage), gas connection (Gas), garbage collection service (Garbage), the population with higher education degrees (Higher Ed.), the percentage of women (Women), and the percentage of children (Children).

### 3.2. Model Fitting and Comparison

A significant shift in both the DIC and WAIC values was evident in the temporal models as opposed to the spatial models (Table 1). Notably, the spatio-temporal models, which integrate a comprehensive set of covariates encompassing socioeconomic, demographic, and transformed environmental predictors, exhibited the most favorable scores in these comparison metrics across all levels and were selected as the best models for multilevel analysis. It is important to note that these models can be compared within the same level but not across different levels.

The fixed effects included in the spatio-temporal models showed variations across different levels of spatial aggregation, as shown in Figure 3. At the level of Comunas, the spatial variables exhibit non-significance, as indicated by the inclusion of zero within the 95% credible intervals for all variables. This lack of significance aligned with the comparable DIC and WAIC values observed for both the temporal and spatio-temporal models in Table 1, thus implying that incorporating spatial variables at this aggregation level does not substantially enhance the model’s explanatory capacity for dengue incidence.

For the intermediate level of Sectores, certain variables such as garbage collection, higher education, and the percentage of children demonstrate significance and displayed inverse correlations with dengue incidence. The negative coefficients suggest that improved access to garbage collection, higher educational attainment, and a larger proportion of children are associated with reduced dengue spread, thus potentially highlighting the impact of enhanced public services and education on disease mitigation.

At the Secciones level, most spatial variables exhibited significance, with garbage collection and the percentage of women showing negative correlations with dengue cases. This inverse relationship suggests that areas with more efficient waste management and a higher proportion of women tend to have lower dengue prevalence. Moreover, factors such as higher population density, gas connections, education levels, and the percentage of children consistently demonstrated negative correlations with dengue cases, thereby echoing the trends observed at the Sectores level and underscoring the influence of these variables on disease incidence.

Finally, at the granular level of Manzanas, sewage connection emerged with a unique positive correlation with dengue cases. Conversely, population density, higher education, and the percentage of children maintained inverse correlations with dengue incidence, thereby aligning with observations made at the Secciones level. This consistency across different levels of spatial analysis suggests that certain factors consistently relate to lower disease incidence despite the finer granularity of data.

Integrating this insight into the preceding analysis highlighted a consistent negative correlation between spatial variables such as higher education, population density, and the percentage of children with dengue incidence across various levels, while the association between garbage collection services and dengue cases appeared less definitive, thus suggesting disparities in waste management service.

Complementing the spatial analysis, Figure 4 introduces lagged temporal predictors through contour plots, which offer insight into the temporal dynamics of the disease. The results underline the cyclical influence of weather patterns on the relative risk (RR) associated with dengue. For instance, the temperature-related metrics, such as mean temperature and the number of days exceeding 32 °C, revealed a lower RR at cooler temperatures and greater lags, where a higher RR transitioned as the temperatures climbed and lag decreased. Conversely, precipitation indicators, such as total precipitation and number of wet days, demonstrated an inverse relationship, with higher precipitation levels correlating with a decreased RR in subsequent periods. Additionally, the NDVI exhibited variability and lacked consistency across different levels of aggregation, thus suggesting complex interactions between vegetation density and disease transmission that warrant further investigation.

### 3.3. Level Comparison

The model’s performance varied across the different levels of spatial aggregation (Figure 5). At the Comunas level, the model achieved a high degree of correlation with the actual dengue case data, although it tended to underestimate case numbers during outbreak peaks—a trend that was particularly pronounced toward the end of 2015. In times of lower disease incidence, such as December 2017 and June 2018, the model also fell short of accurately capturing the case numbers. Despite these limitations, it performed commendably in periods of low case counts, notably throughout most of 2017 and the early months of 2018.

Moving to the Sectores level, the model had a propensity to overestimate the number of dengue cases, with this trend being especially evident in January 2015 and January 2016. This tendency for overestimation continued through the endemic years of 2017 and 2018. Moreover, there was a noticeable misalignment in the timing of the expected outbreaks compared to the actual data, thus highlighting a phase discrepancy between the model fitting and observed case trends.

Similarly, at Secciones level, there was a need for phase alignment. Still, the model demonstrated an accurate fit during the epidemic periods, both in terms of case count and pattern, as seen between September 2015 and January 2016. This level also accurately captured smaller peaks during the endemic periods, such as December 2016 and June 2018. However, during periods characterized by low dengue incidence, such as the late 2017 to early 2018 timeframe, the model tended to overestimate the number of cases, thus indicating a challenge in accurately modelling low incidence rates.

At the most granular level of spatial aggregation, i.e., Manzanas, the model’s fit showed the greatest fluctuation among all the levels. While it aligned more closely with the observed data during the epidemic periods, similar to Secciones, its performance was significantly less accurate during the endemic years, thus indicating a disparity in the model fit across different periods.

The differences in model performance underscored the challenges in capturing the complex dynamics of dengue transmission, which varied temporally during the epidemic and endemic cycles and spatially at different levels of urban granularity. These insights emphasize the need for models that can adjust to both the scale of analysis and the fluctuating nature of disease transmission, thereby highlighting the intricate balance between spatial resolution and accuracy in epidemiological modeling.

A lower RMSE was found for the Comunas level since it provided the best overall fit (Table 2); however, the level of Secciones provided a lower RMSE than the Manzanas and Sectores levels, which might be due to the better fit exhibited during epidemic seasons.

## 4. Discussion

The GWR analysis showed variation in the spatial covariates among the levels of aggregation. Socioeconomic predictors were mostly significant at the lowest level, i.e., Manzanas, while demographic predictors were also significant at intermediate levels like Secciones or Sectores. Only two spatial predictors were significant at the Comunas level. The GWR analysis solely identified significant spatial variables and overlooked non-linear interaction, which is crucial for understanding the endemic–epidemic patterns of dengue. This limitation may result in models that only partially capture the disease’s dynamics.

Our study shows that the spatio-temporal models, which integrate spatial and temporal variability, generally offered a better fit, and they were selected for further analysis at all levels of aggregation. However, the improvement in the spatio-temporal models at levels like Comunas and Sectores was minimal, thus indicating that the added complexity of spatio-temporal models may only sometimes lead to a significantly improved fit. This underscores the importance of careful consideration when increasing model complexity and highlights the need to balance detailed spatio-temporal dynamics with model simplicity.

The significance of the spatial predictors on dengue incidence varied markedly across the different levels of spatial aggregation. At the most aggregated levels, namely Comunas and Sectores, many of the variables were found to be non-significant, possibly due to the homogeneity within these broader spatial categories. The lack of variability within the spatial covariates at these levels leads to a limited ability to discern the significant impacts on dengue cases, as observed in the narrow covariate ranges detailed in Appendix A. In contrast, human population density was notably significant at the more granular Secciones and Manzanas levels, and inversely correlated with dengue incidence. However, this counterintuitive finding is supported by previous studies, which have suggested that higher human population densities may not favor mosquito breeding, particularly if the densely populated areas have sufficient sanitation and utility services [51,52].

Socioeconomic variables, such as sewage, gas connection, and garbage collection service exhibit diverse correlations with dengue cases. Sewage connection is positively correlated and attributed to urban infrastructure and vector ecology (adaptation to breed in manmade environments), including improperly designed systems creating mosquito breeding sites [53,54,55,56]. Conversely, areas with higher gas connection rates tend to have lower dengue incidence, thus reflecting socioeconomic status. The relationship with garbage collection services varies; a positive correlation at the Secciones level and an inverse relationship at the Sectores level suggests that complex dynamics are influenced by local practices and infrastructure [57,58].

The observed variability among socioeconomic variables underscores the intricate interplay of individual and collective dynamics, which sometimes result in counterintuitive outcomes. Nevertheless, these variables offer valuable insights into the internal dynamics of spatial distribution, thereby exhibiting distinct characteristics across different levels of spatial aggregation.

Demographic predictors, including higher education, the percentage of children, and the percentage of women, demonstrated a consistent pattern across the spatial levels. Higher educational attainment in populations may lead to increased implementation of disease prevention measures, potentially reducing breeding sites and subsequent dengue cases [59,60]. The inverse correlation with the percentage of children may reflect the demographic profile of dengue cases during specific outbreaks, with adults and young adults being more frequently affected. Lastly, the percentage of women may reflect the broader demographic composition of the city and the roles women play in household management and potential exposure to mosquito breeding sites.

Variations in the correlation between spatial variables and dengue incidence across different spatial aggregation levels indicate diverse roles of socioeconomic and demographic factors in disease transmission dynamics. While broader scales may obscure these factors amidst other influences, finer scales highlight their heterogeneity, thereby allowing for a more precise understanding of their impact on dengue transmission. This underscores the complexity of disease incidence modeling and emphasizes the importance of considering scale when interpreting variable influences.

Despite the consistent trends in the temporal variables across all spatial aggregation levels, subtle variations in RR suggest differing degrees of correlation with dengue cases. Temperature and precipitation influence were relatively consistent across the city’s spatial structures, thereby resulting in similar impacts on RR regardless of aggregation level. Our results show an association between the mean temperature and increased RR during high-temperature periods (with a zero-month lag), and elevated disease risk due to total precipitation at a three-to-six-month lag across the Comunas, Sectores, and Secciones levels, which underscores the influence of weather patterns on dengue transmission in urban areas. This relationship is consistent with regional climate patterns, where cooler seasons precede warmer periods by approximately four to six months [61]. This pattern is further intensified by ENSO phenomena, such as the El Niño events of 2015 and 2016. These extreme weather events, characterized by higher temperatures and drought, impact mosquito breeding and human access to utilities in vulnerable areas, thus leading to community adjustments in water management practices and influencing dengue transmission dynamics [27,62].

The days over 32 °C exhibited a comparable trend with the mean temperature, where a higher frequency of hot days at shorter lags was linked to an increased RR. The same relationship was displayed by the number of wet days and total precipitation, with increased RR at two- and six-month lags, reflecting the bimodal rainy seasons. Finally, the NDVI showed irregular patterns, with minimal variation at broader levels like Comunas and Sectores due to its non-seasonal nature.

While the temporal patterns held consistently across the spatial aggregation levels, the changes in RR for each covariate varied, thus suggesting differing strengths of correlation with dengue incidence. This variance could stem from how dengue cases are distributed across each spatial level and how spatial covariates account for the observed effects at more granular levels.

This study highlights the importance of considering temporal and spatial variables in understanding dengue dynamics. While temporal variables play a significant role, spatial covariates at finer levels of aggregation are also crucial for a nuanced understanding of dengue transmission. The analysis suggests that the Comunas level model provides the best overall fit for the city, with the Secciones model closely following. However, the Manzanas level model performed weaker due to extreme case dispersion. Despite these dispersion issues, intermediate aggregation levels like Secciones reveal discernible links with socioeconomic and demographic variables, thereby aiding in understanding local patterns for targeted interventions. This underscores the importance of considering macro and micro-level factors in epidemiological modeling and intervention planning to tailor public health strategies and reduce disease prevalence effectively.

The accuracy of dengue case reporting relies on local population engagement, which is often hampered by underreporting due to symptom recognition without seeking formal diagnosis and the prevalence of asymptomatic cases [63]. This underreporting significantly impacts case count accuracy, thus hindering model precision [64]. Census data limitations are apparent, with data available only for 2018, assuming that socio-economic and demographic variables remained unchanged over six years, and the neglect of potential variations.

Modeling efforts focusing on endemic or epidemic periods may offer immediate insights into socio-economic and demographic influences on disease patterns but may overlook long-term effects. In addition, detailed research at lower observational levels is needed to address data scarcity and the influence of local entomological and virological factors on disease dynamics, which are not currently available for the whole city.

## Figures and Tables

**Figure 1 viruses-16-00906-f001:**
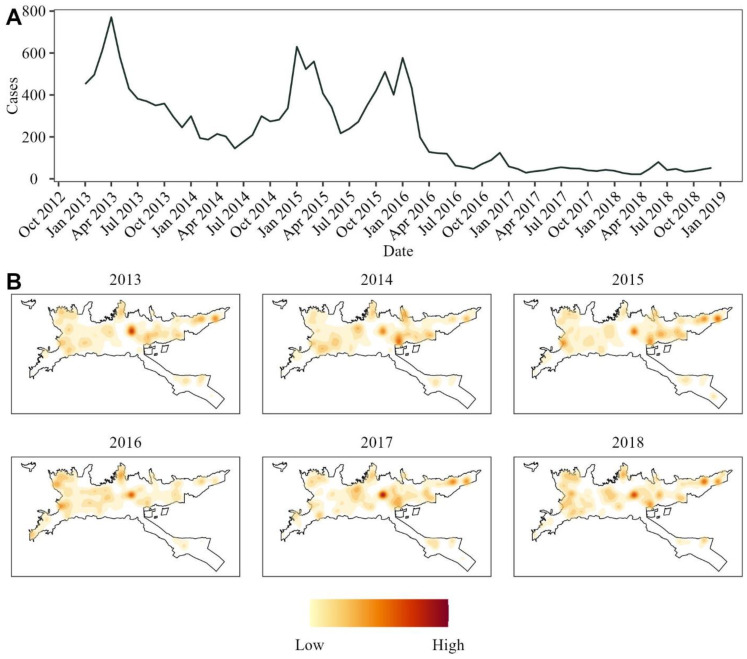
Human dengue cases reported from 2013 to 2018 in Ibagué. (**A**) Time series of the aggregated cases for the whole city. (**B**) Kernel density estimation for the georeferenced cases for each year of the studied period.

**Figure 2 viruses-16-00906-f002:**
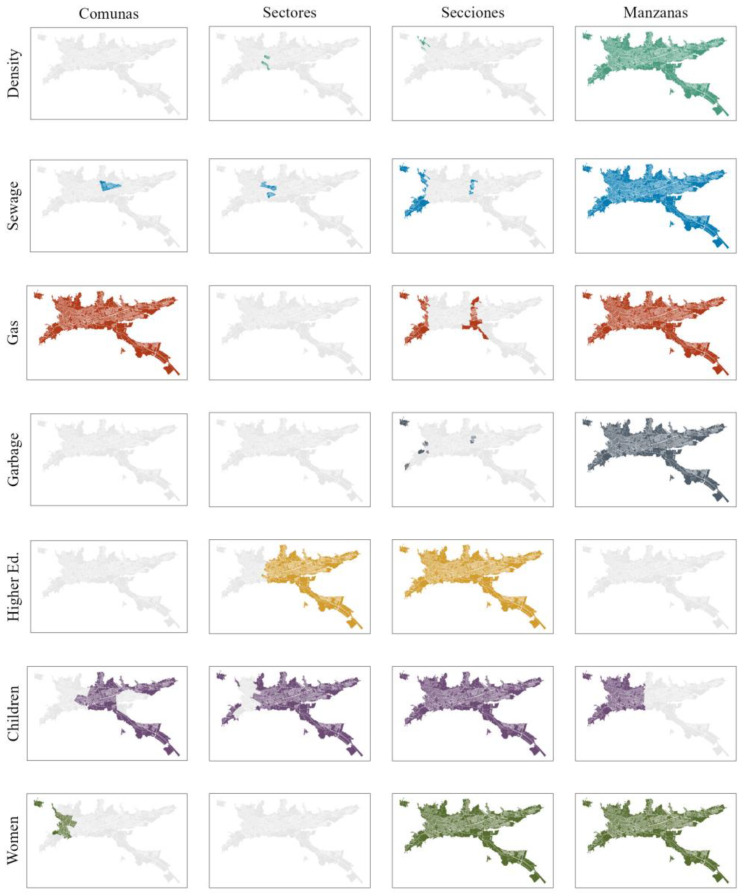
The GWR significance for individual spatial features at each level. The columns represent the different levels, while rows are the significant variables. Colored areas indicate the statistical significance in the local regressions.

**Figure 3 viruses-16-00906-f003:**
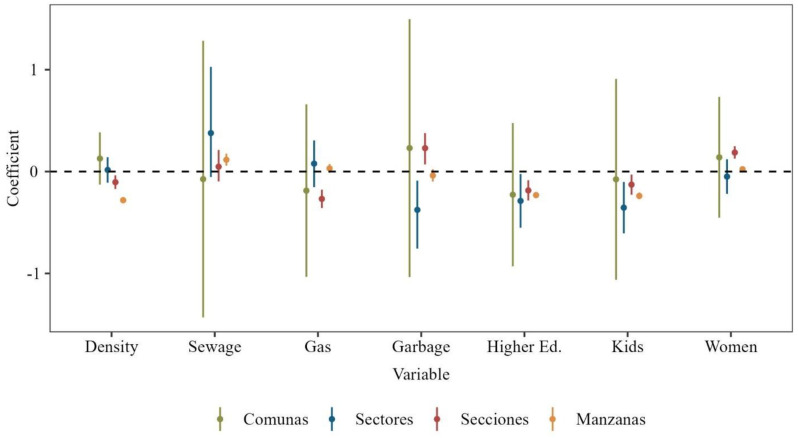
Results for the spatial fixed effects in the spatio-temporal fitted models. The points represent the median of the marginal posterior distributions, and the lines draw the 95% credible interval for each level.

**Figure 4 viruses-16-00906-f004:**
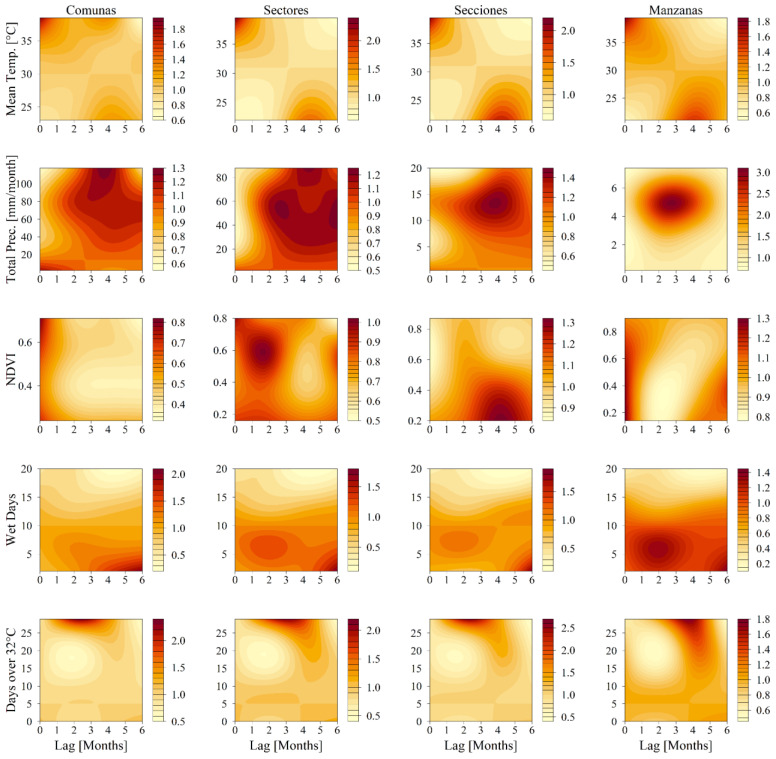
Contour plots for the results of the temporal variables showing the relative risk of human dengue cases from exposure and different time lags. The relative risk was calculated from the baseline observations for each covariate as follows: mean temperature risk was calculated relative to 22 °C; total precipitation was determined relative to 0 mm/month; and the NDVI, wet days, and days over 32 °C were calculated relative to 0.

**Figure 5 viruses-16-00906-f005:**
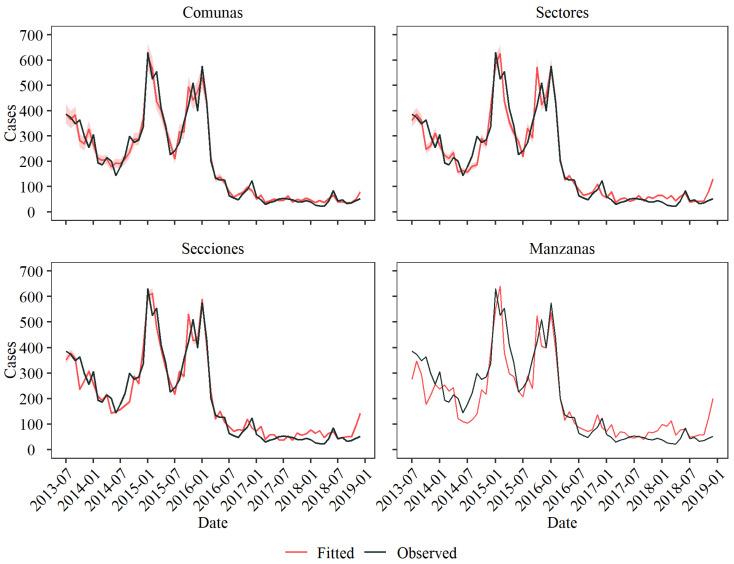
The aggregated fitted values from the marginal posterior distributions compared to the observed dengue cases in the city for the studied period. Fitted values are plotted with a 95% credible interval at each level.

**Table 1 viruses-16-00906-t001:** Model comparison using DIC and WAIC at each level of spatial aggregation.

		Model		
Level	Metric	Spatial	Temporal	Spatio-Temporal
Comunas	DIC	5345	4861	4860
	WAIC	5358	4863	4863
Sectores	DIC	14,646	13,874	13,871
	WAIC	14,656	13,874	13,869
Secciones	DIC	33,942	32,206	32,163
	WAIC	33,871	32,185	32,139
Manzanas	DIC	89,900	84,728	84,051
	WAIC	89,845	84,667	83,988

**Table 2 viruses-16-00906-t002:** The root mean square error (RMSE) for each level’s posterior marginals.

	Comunas	Sectores	Secciones	Manzanas
RMSE	32.69	45.80	42.34	66.63

## Data Availability

All the data and code are available at https://github.com/jd-otero/dengue_ibague.

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
