# Peer review of "Exploring Dengue Dynamics: A Multi-Scale Analysis of Spatio-Temporal Trends in Ibagué, Colombia"

_viruses, 2024, doi:10.3390/v16060906_

Round 1

Reviewer 1 Report

Comments and Suggestions for Authors

The study provides valuable insights into the complex relationships between dengue incidence and spatial-temporal factors, highlighting the importance of considering various factors at different scales. The methodology employed, including GWR and INLA, demonstrates a robust approach to analyzing and understanding the factors influencing dengue transmission. The entire study is logically clear, and I support the publication of this article after minor revisions. Some key minor concerns include:

1Why does the GWR significance in Figure 2 show significant differences for different regional divisions? For example, regarding the variable "children," the results obtained from using larger "comunas" divisions show significant areas on the left side of the region, while using the smallest "manzanas" divisions show significant areas on the right side. Does this indicate that these significances are not significant enough? From Figure 3, it can also be seen that many coefficient values are close to 0.

2The link to the GitHub website at the end cannot be opened.

3One last point I would like to mention is that personally, I do not favor this purely mathematical fitting process. I have great respect for the mathematical fitting method like INLA; this type of fitting will definitely bring good fitting effects when enough parameters are added, but it does not possess predictive ability for the future. To increase the model's feasibility, readers naturally hope to see more analysis in the discussion section regarding the physical relationship between independent and dependent variables. For example, it is easy to understand from Figure 3 that there is a certain positive correlation between sewage and dengue incidence, but most people find it difficult to understand why the percentage of children shows a negative correlation with the total incidence of dengue. Since we all know that many children have not been infected with dengue, they belong to the susceptible population for dengue. When the model's correlation results contradict this physical principle, you need to provide appropriate explanations for these results.

Author Response

Thank you very much for your comments. We attach a document where we have addressed the comments and mention where in the revised manuscript they are considered.

Reviewer 2 Report

Comments and Suggestions for Authors

The authors have presented a very interesting and timely study, especially given the increasing global burden of dengue. There is an urgent need to enhance interventions and uncover potential risk factors to prevent future dengue outbreaks. This study highlights the need of integrating both demographic, socioeconomic, and environmental factors into disease risk models which is crucial for producing more accurate and practical outcomes that can aid public health officials in their decision-making processes. Moreover, this study demonstrates the significance of conducting spatial analyses and disease risk modelling not only on a large scale but also more localized level, which could provide novel insights into understanding dengue dynamics.

However, there are some critical comments to be considered. These include:

·        Providing an inset map of Colombia and neighboring countries alongside the study area map would offer readers a clearer geographical context.

·        Authors refer to the previous study on Aedes mosquitoes from the same city (ref nrs 8 and 9). Because information of vector species distribution is essential on estimating the dengue disease risk, would it be possible to include such factor to the models used in this study? If, for some reason, such information cannot be included, the lack of vector species data should be clearly discussed in Discussion section.

·        The Introduction section contains overly detailed descriptions of other studies. It would be beneficial to condense the text significantly.

·        Ensure that each section (Introduction, Material & Methods, Results, Discussion) includes only information relevant to that section. For example, issues presented in the Introduction that belong to the Methods or Discussion section should be appropriately relocated.

·        Clear and concise Results should be provided.

·        Authors present figure and table captions in the text. The information from the figures/tables should be more discreetly embedded in the text.

Abstract:

Row 17:  Can you provide a more precise definition of Ibague? Is it a city or a province and where in Colombia is it located?

Row 19: Mentioning names of spatial aggregation levels here is unnecessary.

Rows 36-37 Please, use either the abbreviation or spelled word for keywords, avoid extra information or words inside the brackets. Keep the phrases concise, with 1-4 words maximum.

Introduction:

Row 40: Dengue fever (DENV) -> DENV is an abbreviation for Dengue virus, not for Dengue fever. Note that virus and disease are distinct. Correct throughout the text.

Rows 40-43: It would be beneficial to include a brief overview of recent dengue outbreaks/epidemics in the study area or in Colombia.

Row 48: Do you mean annual average temperature?

Row 48: Italicize "Aedes" and specify the species, as not all Aedes species can transmit the dengue virus.

Row 50: Introduce dengue virus transmission briefly, explaining how the virus is transmitted.

Rows 56- 99: Condense the description of other studies and their findings to a maximum of two paragraphs. Also, the massive presentation of the used methods in other studies are not essential in Introduction section (if you´d like to compare methods, include that section to Discussion), please mention only few methods if necessary and condense. Focus on key factors affecting dengue occurrence/incidence, presenting only a few key findings from other studies and referring to them after the sentence for clarity.

Methods:

Row 125: This section would need an inset map including Colombia + neighboring countries with Ibague and the other map which shows the study area as presented in Supplementary materials to provide readers from outside Colombia with geographical context.

Row 128: Describe these more precisely and provide any English terms, along with approximate areas (m2 or Km2) if available.

Rows 130-131: Specify how the original predictor data was chosen, e.g. based on literature?

Row 145: “17,707 DENV” -> I assume that these are not number of viruses but disease cases. Please modify.

Row 147: “the evolution of the virus” -> please correct. The figure shows monthly distribution of human dengue cases not the evolution of the virus..

Row 148: Ensure numbers do not start sentences.

Row 154: Please correct: “Human dengue cases”

Row 155: Ensure Figure B is introduced in the preceding paragraphs.

Row 168: Use lower case for “wavelet coherence”

Row 169: Modify "DENV cases" to "dengue cases" or "human dengue cases" throughout the manuscript.

Row 183: Delete the comma.

Results:

Row 227: Use "The GWR analysis," as this is already introduced in the Introduction.

Rows 234-239: Modify as these sound more like figure captions; "Visualization aids..." is not part of the results but rather methods. Refer to Figure 2 after "The GWR analysis revealed..."

Rows 241-243: This sound like table captions rather than text in the manuscript. Only refer to Table 1 after the sentence "A significant shift..."

Rows 256-258: Similar to earlier comments, these sound like figure captions.

Rows 264-265: If I understood right, results refers to Figure 2? In this case, there are statistically significant variables with e.g. women, children and gas? If I understood wrong, please add ref to the right figure.

Row 269: dengue incidence -> modify throughout the manuscript

Row 270: If I understood right, results refers to Figure 2? If this refers to Figure 2, no statistical significance with garbage? but with sewage and density? If I understood wrong, please add ref to the right figure.

Rows 313-314: Again this sounds like Figure caption. Please modify.

Row 336: Modify "generally low DENV activity" to "e.g., low dengue incidence."

Row 350: Unnecessary. Just refer to Table 1 after the sentence “Lower RMSE…”

Discussion:

Row 358: Use "significant."

Row 372: highest of what? Please rephrase/change the word.

Row 376: Use "human population density."

Row 378: Change "That" to lowercase.

Row 379: Change to "population densities."

Row 456: Ensure this sentence is linked to the preceding one (Row 455).

Supplementary material:

Table S1: Categorize variables into socioeconomic, demographic, and environmental sections, along with spatial and temporal sections. Specify the availability (e.g. MODIS available online at…) and resolution of data.

Comments on the Quality of English Language

The manuscript needs extensive revision for language and grammar. Proofreading by a native English speaker would be beneficial.

Author Response

Thank you very much for your comments. We attach a file with the respective responses, and where in the revised manuscript they are addressed
